# Factors Associated with 90-Day Mortality in Invasively Ventilated Patients with COVID-19 in Marseille, France

**DOI:** 10.3390/jcm10235650

**Published:** 2021-11-30

**Authors:** Maxime Volff, David Tonon, Youri Bommel, Noémie Peres, David Lagier, Geoffray Agard, Alexis Jacquier, Axel Bartoli, Julien Carvelli, Howard Max, Pierre Simeone, Valery Blasco, Bruno Pastene, Anderson Loundou, Laurent Boyer, Marc Leone, Lionel Velly, Jeremy Bourenne, Salah Boussen, Mickaël Bobot, Nicolas Bruder

**Affiliations:** 1Department of Anaesthesiology and Critical Care Medicine, University Hospital Timone, AP-HM, Aix-Marseille University, CEDEX 05, 13385 Marseille, France; davidtonon@ap-hm.fr (D.T.); youribommel@ap-hm.fr (Y.B.); noemieperes@ap-hm.fr (N.P.); davidlagier@ap-hm.fr (D.L.); geoffrayagard@ap-hm.fr (G.A.); howardmax@ap-hm.fr (H.M.); pierresimeone@ap-hm.fr (P.S.); lionelvelly@ap-hm.fr (L.V.); salahboussen@ap-hm.fr (S.B.); nicolasbruder@ap-hm.fr (N.B.); 2Department of Radiology and Cardiovascular Imaging, UMR 7339, CNRS, CRMBM CEMEREM (Centre de Résonance Magnétique Biologique et Médicale-Centre d’Exploration Métaboliques par Résonance Magnétique), Aix-Marseille University, CEDEX 05, 13385 Marseille, France; alexisjacquier@ap-hm.fr (A.J.); axelbartoli@ap-hm.fr (A.B.); 3Emergency and Critical Care Medicine, University Hospital Timone, AP-HM, Aix-Marseille University, CEDEX 05, 13385 Marseille, France; juliencarvelli@ap-hm.fr (J.C.); jeremybourenne@ap-hm.fr (J.B.); 4CNRS, Institut des Neurosciences de la Timone, UMR 7289, CEDEX 05, 13385 Marseille, France; 5Réanimation Polyvalente des Pathologies du Foie, AP-HM, CEDEX 05, 13385 Marseille, France; valeryblasco@ap-hm.fr; 6Service Anesthésie et Réanimation, Hôpital Nord, AP-HM, Aix-Marseille University, CEDEX 20, 13915 Marseille, France; brunopastene@ap-hm.fr (B.P.); marcleone@ap-hm.fr (M.L.); 7Laboratoire de Santé Publique, Unité de Recherche EA 3279, Faculté de Médecine de Marseille, Aix-Marseille University, CEDEX 05, 13385 Marseille, France; andersonloundou@ap-hm.fr; 8CEReSS-Health Service Research and Quality of Life Center, Aix-Marseille University, CEDEX 05, 13385 Marseille, France; laurentboyer@ap-hm.fr; 9Department of Medical Information and Public Health, University Hospital Timone, APHM, Aix-Marseille University, CEDEX 05, 13385 Marseille, France; 10Department of Epidemiology and Health Economics, AP-HM, CEDEX 05, 13385 Marseille, France; 11LBA, UMRT 24, IFSTTAR, Aix-Marseille University, Boulevard Pierre Dramard, CEDEX 20, 13916 Marseille, France; 12Service de Médecine Intensive Réanimation, AP-HM, Hôpital Nord, CEDEX 20, 13915 Marseille, France; mickaebobot@ap-hm.fr; 13INSERM 1263, INRAE 1260, C2VN, Aix-Marseille University, CEDEX 05, 13385 Marseille, France

**Keywords:** Covid-19, SARS-CoV-2, intensive care unit, acute respiratory distress syndrome, mechanical ventilation, prognostic factors

## Abstract

Objectives: To describe clinical characteristics and management of intensive care units (ICU) patients with laboratory-confirmed COVID-19 and to determine 90-day mortality after ICU admission and associated risk factors. Methods: This observational retrospective study was conducted in six intensive care units (ICUs) in three university hospitals in Marseille, France. Between 10 March and 10 May 2020, all adult patients admitted in ICU with laboratory-confirmed SARS-CoV-2 and respiratory failure were eligible for inclusion. The statistical analysis was focused on the mechanically ventilated patients. The primary outcome was the 90-day mortality after ICU admission. Results: Included in the study were 172 patients with COVID-19 related respiratory failure, 117 of whom (67%) received invasive mechanical ventilation. 90-day mortality of the invasively ventilated patients was 27.4%. Median duration of ventilation and median length of stay in ICU for these patients were 20 (9–33) days and 29 (17–46) days. Mortality increased with the severity of ARDS at ICU admission. After multivariable analysis was carried out, risk factors associated with 90-day mortality were age, elevated Charlson comorbidity index, chronic statins intake and occurrence of an arterial thrombosis. Conclusion: In this cohort, age and number of comorbidities were the main predictors of mortality in invasively ventilated patients. The only modifiable factor associated with mortality in multivariate analysis was arterial thrombosis.

## 1. Introduction

Coronavirus disease (COVID-19) outbreak caused by severe acute respiratory syndrome coronavirus 2 (SARS-CoV-2) has spread worldwide from China. It was declared pandemic by World Health Organization in March 2020 [1]. Despite all efforts its propagation still remains difficult to manage in Western countries, placing constant pressure on healthcare systems by an increasing need for intensive care units (ICUs) beds and prolonged hospitalisations. As of 1 June 2021, 5,677,172 confirmed cases and 109,662 related-deaths were reported in France [2]. To date, several publications [3,4,5,6,7,8] have given a consistent description of the critically ill patients with COVID-19. Most have reported a 28-day mortality ranging between 28 and 44% with patients still receiving ICU support at the point of publication. To our knowledge, only two European publications reported a 3-months mortality. The first is the French prospective study COVID-ICU [9], which described a cohort of 4244 patients admitted in ICU with a 90-day mortality of 31%. The second is the Dutch retrospective study ProVENT-COVID [10], which reported a 43% mortality rate at day 90 in a cohort of 533 patients who all received invasive mechanical ventilation (IMV). The aim of our multicentre study was to describe the characteristics and management of the 172 critically ill patients admitted in our institution (Assistance Publique—Hôpitaux de Marseille, AP-HM, Marseille, France) and to evaluate risk factors associated with 90-day mortality. Particular attention was given to the group of invasively ventilated patients to allow comparison with other works.

## 2. Materials and Methods

### 2.1. Study Design

In this observational retrospective study data were collected from all consecutive adult patients with SARS-CoV-2 admitted to six ICUs of three university hospitals in Marseille (Hôpital de la Timone, Hôpital Nord de Marseille, Hôpital de la Conception, Marseille, France) in a single institution (Assistance Publique des Hôpitaux de Marseille, AP-HM—Aix Marseille University, Marseille, France) during the first wave of the outbreak from 10 March 2020 until 10 May 2020. The SARS-CoV-2 positive diagnosis was defined as a positive result of real-time reverse transcriptase-polymerase chain reaction (PCR) assay of nasal or pharyngeal swabs. Some patients were excluded from analysis, including patients with unconfirmed positive PCR, patients without respiratory symptom and those with viral confirmation concomitant to another organ failure.

### 2.2. Data Collection

Data were obtained retrospectively for each patient from their electronic charts. We collected the baseline characteristics including age, sex, body mass index (BMI) > 25 kg/m^2^, comorbidities and long-term medications. Metabolic syndrome was confirmed using the International Diabetes Federation definition [11]. ABO blood group was reported using electronic records of our institutional blood bank. To assess the severity, several scoring systems were used based on the data collected during the first 24 h after ICU admission: SAPS II (Simplified Acute Physiology Score II), SOFA (Sequential Organ Failure Assessment), APACHE II (Acute Physiology and Chronic Health Evaluation II), Charlson comorbidity index, NEWS and modified NEWS scores [12], MuLBSTA score, Murray score, CURB-65 score and ROX index. When patients underwent non-invasive ventilation, an oxygen flow—FiO_2_ conversion table was used to estimate the PaO_2_:FiO_2_ ratio (see Appendix A). Regarding the laboratory results, only worst values within first 48 h after ICU admission were considered for statistical analysis. We reported analysis of the first chest computed tomography (CT) scan performed after hospital admission, if one exists, based on automated volumetry. Severity of disease based on CT scan was then graded as normal, minimal, intermediate or severe by our radiologists, based on the number of affected pulmonary segments (See Appendix B).

The patient management, including supportive measures and pharmacologic agents, was performed at the discretion of the treating physicians, including decision of tracheal intubation. Respiratory support devices, mechanical ventilation settings and pressure levels in the first 24 h were recorded. For the patients undergoing IMV, Day-1 PaO_2_:FiO_2_ ratio and its corresponding acute respiratory distress syndrome (ADRS) severity based on the Berlin definition were reported [13]. Mechanical power (J/min) was calculated using following formula: 0.098 × tidal volume × respiratory rate × (peak pressure—0.5 × driving pressure) [14]. Driving pressure was defined as plateau pressure minus positive end-expiratory pressure (PEEP).

Significative events and complications during ICU stay such as thrombosis or nosocomial infections were reported. Ventilator-associated pneumonia was only considered if clinical suspicion was associated with microbiological documentation. Otherwise, all clinical and radiological criteria were required [15]. Sepsis-3 criteria were used for sepsis shock authentication [16]. Withholding an invasive ventilation for ethical reasons formally documented in the patient chart was reported. Number of daily available ICU beds and occupation rates were obtained by the administration of our local institution.

Finally, we considered date of symptoms onset, date of admission to hospital and ICU, date of death or hospital discharge and vital status at hospital discharge and after 3 months for evaluation. For this last information, we contacted hospitals and rehabilitation centres by phone if the patients had already left our institution.

### 2.3. Statistical Analysis

Continuous variables are expressed as mean ± standard deviation or as median with interquartile range (Q1, Q3), and categorical variables are reported as count and percentages. Comparisons of means values between two groups were performed using student *t*-test or Mann–Whitney U. Comparisons of percentages were performed using Chi-square test or (Fisher’s exact test, as appropriate). The overall survival (OS) was defined as the time from the date of ICU admission to date of death. In order to identify predictive factors of death, univariate and multivariate survival analyses were performed using the Cox proportional-hazards model. Multivariate analysis included variables that were statistically significant in the univariate analysis and takes into account multiple comparisons with an FDR analysis. The results are reported as hazard-ratios with 95% confidence intervals. All statistical tests were two-sided and the threshold for statistical significance was *p* < 0.05. Statistical analysis was performed using PASW Statistics version 17.02 (IBM SPSS Inc., Chicago, IL, USA).

## 3. Results

### 3.1. Enrolled Patients and Characteristics

Between 10 March 2020 and 10 May 2020, 172 patients were admitted in ICU for a pneumonia with laboratory-confirmed SARS-CoV-2 infection. The flow chart of the study inclusions is reported in Figure 1. Their complete characteristics and outcomes are reported in Appendix A. Evolution of daily inpatient prevalence and available ICU capacity during this period is reported in Figure 2.

117 patients (67%) received IMV. Baseline characteristics are reported in Table 1. At ICU admission median age of the ventilated patients was 63 (56–72) years, with 88 men (75.2%) and 29 women (24.8%). The most frequent reported comorbidities were hypertension 68/117 (58.1%), BMI > 25 kg/m^2^ 73/117 (62.4%), diabetes 24/117 (20.5%) and immunodeficiency 24/117 (20.5%). The average Charlson comorbidity index was 3 (2–4). Their median SAPS II and SOFA score in the first 24h after ICU admission were 34 (27–40) and 5 (3–7), respectively. A chest computed tomography in the first days after hospitalisation was available for 84/117 patients (72%). Based on automated volumetry, the median volume of lung lesions was 31.8% (15.6–46.6) of the parenchyma, mostly ground glass. Lymphopenia was commonly observed, with a median value of 0.69 (0.5–0.95) × 10^9^/L. The inflammatory syndrome was characterised by elevated values for CRP and ferritin with respective medians of 201 mg/L (126–302) and 1418 ng/mL (968–2321).

### 3.2. Severity of ARDS and Respiratory Support

High flow oxygen and non-invasive ventilation before intubation were used for 69/117 (59%) and 16/117 (13.7%) patients, respectively. In these invasively ventilated patients, 19/117 (16.2%) suffered from mild, 60/117 (51.3%) from moderate and 24/117 (20.5%) from severe ARDS. Veno-venous Extra Corporeal Membrane Oxygenation (ECMO) was provided to 25/117 (19.7%) patients with a median duration of 13.5 (10–21.5) days.

### 3.3. Complications and Outcomes

Complications and outcomes according to the day-90 survival status are presented in Table 2. Ventilator-associated pneumonia occurred in 71/117 (60.7%). Venous thrombosis (including patients suffering from pulmonary embolism) were diagnosed in 35/117 (29.9%) while arterial thrombosis were diagnosed in 10/117 (8.5%). Of note, a severe bleeding event was reported in 25 patients (21.5%).

The 90-day mortality rate was 27.4% for the patients requiring IMV. Among the patients who received invasive or non-invasive ventilation on the day of ICU admission, the day-90 mortality rate increased with the severity of ARDS at ICU admission (13.5%, 23.5% and 28.6% for mild, moderate and severe ARDS, respectively). Median duration of ventilation and median length of stay in ICU for intubated patients were 20 (9–33) days and 29 (17–46) days, respectively. Multivariable analysis is shown in Table 3. Age, Charlson index, chronic statins treatment and arterial thrombosis were associated with increased risk of 90-day mortality in the group of mechanically ventilated patients (analysis of survival according to age categories is reported in Appendix A).

## 4. Discussion

### 4.1. Mortality

In this cohort of 172 critically ill patients with COVID-19, overall 28-day and 90-day mortality were 15.7% and 21.5%, respectively. This is noticeably lower than in previous reports [3,4,5,6,7,8,9,10,17]. As a major risk factor for mortality, the various proportion of patients undergoing IMV in each of these cohorts seems to be the main determinant of these heterogeneous outcomes. Indeed, an observational study in Vancouver [8] with a similar number of intubated patients reported a comparable mortality in ICU. If we focus on the day-90 outcome, our observation is consistent with the mortality of 31% reported in the European prospective study COVID-ICU [9] where 80% of patients had undergone IMV. Nevertheless, even the number of deaths in the subgroups of invasively ventilated patients varies widely between these studies, ranging from 30% to 96.8%. In our cohort, corresponding 28-day and 90-day mortality was 18.8% and 27.4%, respectively. Lower mortality and later deaths led in our cohort to longer lengths of ICU and hospital stays with 29 (17–46) and 37 (24–53) days for ventilated patients respectively. Median duration of IMV was also longer than in other studies and that traditionally observed in non-COVID ARDS [18]. This can create a vicious circle in overwhelmed hospitals: if surviving from COVID-19 needs time, physicians may be faced with difficult limitations to advance life support in patients with long invasive ventilation.

### 4.2. Risk Factors

Baseline characteristics of the mechanically ventilated patients in our cohort are similar to those reported elsewhere. Age, number of comorbidities and dyslipidaemia authenticated by statins intake were associated with 90-day mortality in multivariate analysis. As previously reported, male gender, hypertension, diabetes or metabolic syndrome are overrepresented in these critically ill patients compared to the general population, but are consistent variables in a multivariate model to predict death. More surprisingly, overweight stands in this cohort as a protective factor for mortality. This result has to be taken with caution due to the absence of stratification on BMI and the potential coexistence of confusion factors. This may be due to an empirical decision for early ICU admission management, showing a greater concern of practitioners for these patients. As well, studies with more complete data for this variable found no effect of obesity on final mortality in ICU [9,10,19,20]. Looking now at initial results of paraclinical exams, we did not find any unexpected parameter associated with mortality; most of them attest to the severity, but are unable to predict good or bad evolution in severe cases. Regarding ABO blood group distribution, we did not find any difference with general population, as it was mentioned is some other publications [21,22,23]. Finally, it has been reported that severity of the extent on initial chest CT could be predictive of a bad evolution [24]. In our cohort, we found an association between damage volumes assessed by automated volumetry or by visual scoring and 90-day mortality only in univariate analysis. Statistical difference on the criteria of total lungs volume shall be ignored, as a quick analysis showed that age was an obvious confusion factor. It is important to note that due to the pathophysiology of the COVID-19 disease and the extensive lesions that occur over time, the time to perform chest CT is probably an important variable to handle. If we examine the prognostic scores, usual critical care score, such as SAPS II, still remained the most accurate to predict final outcome. Respiratory scores calculated on initial parameters were not relevant to estimate probability of death. Lastly, number of arterial thrombotic events was significantly higher in non-survivors despite similar anticoagulation therapy, but it occurred mostly in already severe cases, including, for example, patients on ECMO, representing 5/10 (50%) cases of arterial thrombosis.

### 4.3. Management

We will now formulate several hypotheses to explain the lower mortality observed in our cohort. First, median values of the different severity of illness scores, including SAPS II, APACHE II or SOFA, are slightly but invariably lower than in the other studies. As our patients were comparable in terms of age distribution and comorbidities, it would mean that we had encountered less severe cases. However, this difference was not found in the group of invasively ventilated patients when regarding PaO_2_/FiO_2_ ratio, ventilation pressure levels or lung compliances recorded after intubation. In our opinion, the main point to highlight is the late date of our first admissions in ICU compared to the rest of France and Europe. Indeed, our first critically ill patient was admitted on 10 March when general containment began in France only seven days later on 17 March. If this measure did have an efficiency, then it certainly benefited us by reducing the surge of patients, maybe allowing admission of less severe cases. Our study confirms that a major increase in ICU beds may be organised without any significant increase in mortality. After reorganisation our institution included up to 238 ICU beds, against 109 ICU beds before the outbreak. Finally, the highest number of COVID inpatients in ICU at the same time was 104, allowed us to admit them all in standard ICU beds (see Figure 2). In addition, early stop of nonurgent surgeries contributed to release the pressure on beds, but also enabled allocation to ICUs of qualified professionals, such as nurses with previous experience in critical care and anaesthesiologists, who are also trained in intensive care medicine in France. Lastly, special efforts were made by the different private hospitals in Marseille, which admitted a large number of patients with COVID-19 in all degrees of severity. As we truly believe that there is a strong correlation between workload in ICUs and global mortality [25], we are convinced that all mentioned measures contributed to the observed outcomes.

The second great advantage of a late epidemy was the possibility we had to learn from the experience formed in the first plagued regions. It consisted mostly of two salient points. First of them was the rapid emerging evidence of an endothelial dysfunction [26,27], with noticeably high number of thrombotic events. On 3 April 2020, the French Society of Anaesthesia and Intensive Care Medecine published guidelines assuming the necessity to treat severe cases with curative anticoagulation [28]. At this time, only 16% of our patients had already left ICU or were deceased, thus the others probably benefitted from this therapy although it is not still clearly established by strong evidence. The second point concerns respiratory support devices. Initially, early intubation was widely practiced in Europe [29], what could have led to a misconception of COVID-19 as a respiratory disease with two profiles [30]. However, this was not reported in later studies [9,10,31]. Observing that the use of high-flow nasal canula had not led to an increase of contaminations in the units that had tried it, we provided therefore this support to almost all the patients who did not request immediate invasive ventilation. We held back intubation until it seems unavoidable, what explains the lower rate of invasively ventilated patients in our cohort. The trend over time observed in the study COVID-ICU [9] confirmed that this rate did not only depend on the ARDS severity but also on the choice of an early or delayed intubation strategy.

The last hypothesis to consider is the strategy of massive testing that was adopted by our institution [32]. This has been reported elsewhere as a protective factor [33]. Early recognition and hospitalisation of severe cases might have prevented some additional admissions of exhausted patients who would have waited too long at home, until the moment where invasive ventilation had become inevitable. Notably, no specific antiviral treatment (hydroxychloroquine, lopinavir/ritonavir, etc.) was associated with lower mortality in our cohort.

### 4.4. Strengths and Limitations

The major strengths of this study are the exhaustive description of our cohort, including original data rarely reported in similar publications up to this day like prognostic scores or chest tomography volumetry, and the 3-months follow-up. Our study included all consecutive patients with COVID-19 related acute respiratory failure. As we were able to maintain usual standard in our critical care practices, due to a contained pressure on our ICUs, we believe that outcomes presented here are much closer to the true clinical course of the disease.

This study has also several limitations. Our cohort is relatively small. Data were collected retrospectively from electronic charts which lead to a significative proportion of missing data or the absence of stratification for some variables. In particular, data concerning mechanical parameters were scarce. Of course, because of the retrospective nature of the study, any relationship between engaged therapies and final vital status should be interpretated with caution. This study was conducted before the results of the RECOVERY trial [34] and then the wide use of dexamethasone in severe COVID-19; therefore, we cannot analyse its effects. Finally, all inclusions were made in the same geographical area, which could limit an extension of the conclusions to other areas.

## 5. Conclusions

In this case series of 172 critically ill patients with COVID-19 in Marseille, France, the 90-day mortality was 21.5% in the whole cohort and 27.4% in the group of mechanically ventilated patients. This is lower than in many previous similar publications and it may reinforce the idea of a correlation between collective capacity to contain the overwhelming aspect of ICUs and final outcomes. This study allowed us to learn lessons for future COVID-19 outbreaks. Age and comorbidities had a major impact on outcome. The only modifiable factor associated with mortality in multivariate analysis was arterial thrombosis. The prevalence of thrombotic events was very high in our cohort. This encourages an empirical strategy of systematic anticoagulation despite no clear evidence of its benefit in randomised trials. It shows also that a safe increase in ICU beds may be anticipated during pandemics with the help of anaesthesiologists and nurses who are not practising ICU care on a regular basis but have an experience in intensive care, supervised by ICU specialists. These observations may also encourage some empirical strategies that have been now widely adopted, such as systematic anticoagulation or delayed intubation, even though randomised studies are needed.

## Figures and Tables

**Figure 1 jcm-10-05650-f001:**
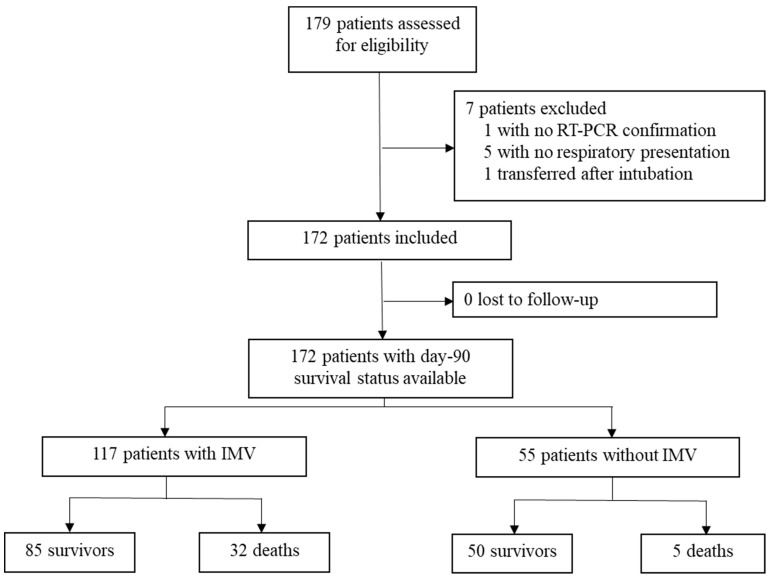
Flow Chart of the population study.

**Figure 2 jcm-10-05650-f002:**
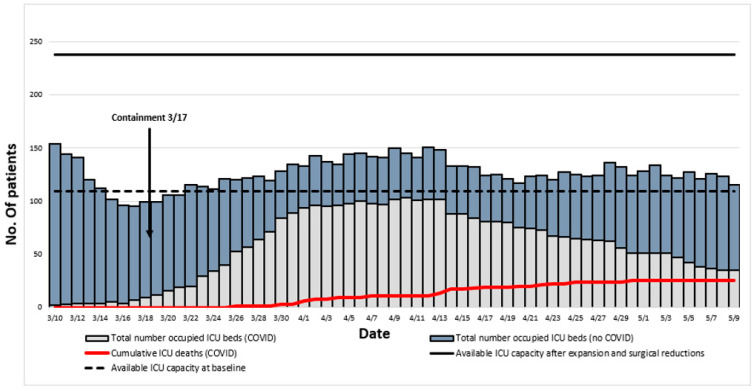
Evolution of daily inpatient prevalence and available ICU capacity.

**Table 1 jcm-10-05650-t001:** Baseline characteristics of the 117 patients on invasive mechanical ventilation according to their 90-day survival status.

	Total (*N* = 117)	Survivors (*N* = 85)	Non-Survivors (*N* = 32)	*p*-Value
Baseline characteristics				
Age	63 (56–72)	61 (54–67)	71.5 (62.75–77)	**<0.001**
Male	88 (75.2)	64 (75.3)	24 (75.0)	0.974
ABO blood group				
A	42 (35.9)	32 (37.6)	10 (31.2)	0.605
B	17 (14.5)	13 (15.3)	4 (12.5)	0.419
O	41 (35)	30 (35.3)	11 (34.4)	0.366
AB	2 (1.7)	1 (1.2)	1 (3.1)	0.218
BMI > 25 kg·m^2^	73 (62.4)	59 (69.4)	14 (43.8)	**0.011**
Hypertension	68 (58.1)	48 (56.5)	20 (62.5)	0.556
Diabetes	24 (20.5)	15 (17.6)	9 (28.1)	0.211
Metabolic syndrome	34 (29.1)	29 (34.1)	5 (15.6)	**0.05**
Chronic respiratory disease	22 (18.8)	15 (17.6)	7 (21.9)	0.602
Chronic kidney disease	11 (9.4)	8 (9.4)	3 (9.4)	0.995
Immunodeficiency	24 (20.5)	15 (17.6)	9 (28.1)	0.211
Statins intake	23 (19.7)	12 (14.1)	11 (34.4)	**0.014**
Systemic steroids intake	9 (7.7)	6 (7.1)	3 (9.4)	0.675
Charlson index	3 (2–4)	2 (1–3)	4 (2–6)	**<0.001**
SAPS II score	34 (27–40)	30.5 (25–38.25)	38 (35–45)	**0.02**
SOFA score	5 (3–7)	4 (3–7)	5 (3–6)	0.966
Total lungs volume on CT (cm^3^)	3257 (2447–4016)	3142 (2285–3804)	3742 (3275–4158)	**0.01**
Lesions/lungs ratio on CT (%)				
Ground glass/lungs ratio	24.8 (12.6–34.4)	26.9 (13.6–36.6)	21.5 (7.1–28.2)	0.862
Condensations/lungs ratio	5.1 (2.1–12.4)	6.6 (3.1–16)	2.1 (0.3–8.2)	0.495
All lesions/lungs ratio	31.8 (15.6–46.6)	34.1 (19.9–49.7)	26.8 (9.2–36.3)	**0.034**
Days from symptoms onset to intubation	9 (6–11.25)	8.5 (6–11)	9 (6–12)	0.3
Days from ICU admission to intubation	0 (0–1)	1 (0–1)	0 (0–1)	0.364
Biology (worst value during first 48H after ICU admission)				
Lymphocyte count (×10^9^/L)	0.69 (0.5–0.95)	0.7 (0.53–1)	0.61 (0.44–0.78)	0.382
Neutrophil to Lymphocyte Ratio	11.1 (8.3–15.5)	10.6 (7.5–15.3)	12.6 (10.5–19.8)	**0.005**
D-Dimers (mg/L)	3.44 (1.64–5)	3.34 (1.52–5)	4 (1.84–5)	0.9
Fibrinogen (g/L)	8.1 (6.9–9.4)	8.1 (7.1–9.5)	8 (6.9–9.4)	0.995
CRP (mg/L)	201(126–302)	179 (120–248)	283 (162–324)	0.29
Creatinine (µmol/L)	85 (67–138)	84 (65–127)	98 (78–153)	**0.048**
LDH (UI/L)	447 (368–535)	435 (347–491)	539 (443–635)	**0.005**
Ferritin (ng/mL)	1418 (968–2321)	1175 (950–1778)	2728 (2342–6049)	**0.021**
Invasive ventilation parameters during first 24 h				
PaO_2_/FiO_2_	130 (100–180)	140 (100–180)	120 (100–160)	0.457
PEEP (cmH_2_O)	12 (10.3–14)	12 (12–14)	12 (10–15)	0.483
Plateau pressure (cm H_2_O)	25 (21.7–28.3)	23.5 (21.0–28.2)	26.0 (23.6–28.8)	0.353
Respiratory compliance (mL/cmH_2_O)	33 (28.2–45)	40 (29–46)	31 (27–40)	**0.018**
Mechanical power (J/min)	15.6 (13.3–19.6)	15.5 (13.5–19.1)	15.9 (12–19.8)	0.836

Results are expressed as *n* (%) or median (25th–75th percentile). Statistical significance in bold.

**Table 2 jcm-10-05650-t002:** Management, complications and outcomes of the 117 patients on invasive mechanical ventilation according to their 90-day survival status.

	Total (*N* = 117)	Survivors (*N* = 85)	Non-Survivors (*N* = 32)	*p*-Value
Management in ICU				
Use of non-invasive ventilation before intubation	16 (13.7)	12 (14.1)	4 (12.5)	0.693
Use of high-flow oxygen before intubation	69 (59)	51 (60)	18 (56.2)	0.589
Neuromuscular blockade	113 (96.6)	82 (96.5)	31 (96.9)	0.923
Prone positioning	99 (84.6)	72 (84.7)	27 (84.4)	0.965
ECMO	23 (19.7)	18 (21.2)	5 (15.6)	0.501
Vasopressors	105 (89.7)	74 (87.1)	31 (96.9)	0.122
Renal replacement therapy	20 (17.1)	12 (14.1)	8 (25)	0.167
Corticosteroids ^a^	23 (19.7)	14 (12.0)	9 (7.7)	0.157
Hydroxychloroquine (10 days) with azithromycin (5 days)	45 (38.5%)	36(42.4)	9 (28.1)	0.159
Remdesivir	0 (0)	0(0)	0 (0)	-
Lopinavir-ritonavir	18 (15.4)	13(15.3)	5 (15.6)	0.965
Complications				
Ventilator associated pneumonia	71 (60.7)	53 (62.4)	18 (56.2)	0.547
Septic shock	47 (40.2)	32 (37.6)	15 (46.9)	0.364
Venous thrombosis or pulmonary embolism	35 (29.9)	29 (34.1)	6 (18.8)	0.106
Arterial thrombosis	10 (8.5)	4 (4.7)	6 (18.8)	0.015
Severe bleeding event	25 (21.4)	17 (20)	8 (25)	0.556
Outcomes				
Duration of ventilation (days)	20 (9–33)	21 (11–34)	18 (6.75–25.25)	**0.06**
Ventilator-free days at d28 (days)	2 (1–7)	4 (1–7)	0 (0–1)	**<0.001**
Length of stay in ICU (days)	29 (17–46)	33 (19–53)	21 (6.75–31.75)	**0.02**
Length of in hospital (days)	37 (24–53)	42 (29–57)	25 (8.75–38.25)	**<0.001**
28-day mortality	21 (17.9)	0 (0)	21 (60)	-

Results are expressed as *n* (%) or median (25th–75th percentile). Statistical significance in bold. ^a^ Irrespective of the dose and the timing.

**Table 3 jcm-10-05650-t003:** Univariate and multivariate Cox regression analysis (*N* = 117).

Associated Factors	Univariate HR (95% CI)	*p*-Value	Multivariate 1 HR (95% CI)	*p*-Value	Multivariate 2 HR (95% CI)	*p*-Value
Age ≥ 65 years	3.27 (1.51–7.08)	**0.007**	4.17 (1.48–11.73)	**0.010**	-	-
Male	0.86 (0.39–1.92)	0.718	-	-	-	-
Charlson Index ≥ 3	5.58 (1.96–15.90)	**0.005**	-	-	3.72 (1.07–12.92)	**0.05**
Arterial thrombosis	2.22 (0.91–5.42)	0.098	3.79 (1.22–11.80)	**0.022**	2.86 (1.00–8.20)	**0.05**
Statins intake	2.55 (1.23–5.31)	**0.020**	3.78 (1.51–9.43)	**0.010**	3.59 (1.44–8.93)	**0.186**

HR (95% CI): Hazard Ratio (95% Confidence Interval). Due to collinearity between age and Charlson Comorbidity index, we performed two different models with the same included variables, but in model 1 with age and model 2 with Charlson Comorbidity index. Statistical significance in bold with an FDR analysis.

## Data Availability

The data presented in this study are available on request from the corresponding author: max-ime.volff@laposte.net.

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
