# Peer review of "Factors Associated with 90-Day Mortality in Invasively Ventilated Patients with COVID-19 in Marseille, France"

_jcm, 2021, doi:10.3390/jcm10235650_

Round 1
Reviewer 1 Report
This study is retrospective analysis of invasively ventilated patients suffering from severe COVID-19. The authors collected and analyzed a huge amount of parameters. Was there any correction for multiple comparisons? Why did you evaluate so many parameters without drawing any conclusion from it? To draw reliable conclusions about the primary endpoint and risk factors for death with this study design the number of included patients is far too small. The authors state that they have evaluated 172 patients, but only 117 are included in the actual analysis. Far more patients over a longer period of time need to be evaluated.
Reviewer 2 Report
In my opinion, the study is well written, and the authors used a suitable methodology. The data allow a partial view of the phenomenon because they refer to a limited sample (as described within the limits of the study). Statistical analysis is well conducted. The only change I suggest: The bibliographic references in the text have been reported individually even when they are multiple, e.g. [1][2][3] must be [1-3]. For that reason, minor revision has needed.Author Response
Please see the attachment

Round 2
Reviewer 1 Report
Thank you for the comprehensive revision of your manuscipt. Most of my comments have been taken into account. However, the abstract needs to be improved. Your manuscript is entitled "Factors associated with 90-day mortality in invasively ventilated patients with COVID-19 in Marseille, France.", but the results section of the abstract does not describe the association of risk factors with mortality. The conclusion section should only refer to results mentioned in the results section.
